# Micronutrient Deficiencies and Determinants Among Pregnant Women and Children in Nigeria: Systematic Review and Meta-Analysis

**DOI:** 10.3390/nu17142338

**Published:** 2025-07-17

**Authors:** Glory Aigbedion, Pei-Ching Tseng, Shuby Puthussery

**Affiliations:** Maternal and Child Health Research Centre, Institute for Health Research, University of Bedfordshire, Park Square, Luton LU1 3JU, UK; glory.aigbedion@beds.ac.uk (G.A.); pei-ching.tseng@beds.ac.uk (P.-C.T.)

**Keywords:** micronutrients, malnutrition, pregnant woman, child, prevalence, risk factors, Nigeria

## Abstract

**Background**: Micronutrient deficiencies, particularly among pregnant women and children under five years old, remain a significant public health challenge in Nigeria. Despite existing policies and programmes, national data on prevalence and risk factors are fragmented. Objective: To synthesise the current evidence on the prevalence of key micronutrient deficiencies and associated risk factors among pregnant women and children under five years old in Nigeria. **Methods**: A systematic review and meta-analysis were conducted using peer-reviewed studies that were published between 2008 and 2024. The databases searched included PubMed, Scopus, and African Journals Online. After screening 1207 studies, 37 studies were included: 27 were conducted among pregnant women and 10 were among children. A meta-analysis was conducted to estimate the anaemia prevalence using a random-effects model. A narrative synthesis was conducted to synthesise evidence on other micronutrients (i.e., magnesium, copper, and vitamins C and E) due to the limited data and risk factors. **Results**: The pooled prevalence of anaemia was 56% among children and 54% among pregnant women. The prevalence of other micronutrient deficiencies varied widely, with a high prevalence of zinc (86.4%), magnesium (94%), and vitamin D (73.3%) deficiencies in certain regions. The identified risk factors included poor dietary diversity, lower socioeconomic status, low maternal education, infection burden, and early or high parity. Most studies were facility-based and sub-national, limiting the generalisability. **Conclusions**: This review highlights a high prevalence of anaemia and micronutrient deficiencies among pregnant women and children in Nigeria. Key risk factors included a poor diet, low maternal education, infections, and reproductive health challenges. Targeted, multisectoral policies are urgently needed to address these gaps and improve health outcomes.

## 1. Introduction

Micronutrients, the essential vitamins and minerals required in small amounts, are vital for optimal physiological functioning and are ideally obtained through a balanced diet [1]. Micronutrient deficiencies occur when an individual lacks or has insufficient amounts of these nutrients due to an inadequate dietary intake, infectious diseases, or impaired absorption of essential nutrients, such as iron, iodine, zinc, and vitamin A, among others [2]. Micronutrient deficiencies remain one of the most widespread forms of malnutrition [3], with estimates ranging from 2 billion to 5 billion people worldwide [4]. In Nigeria, micronutrient deficiencies remain a significant public health problem. The 2018 Nigeria Demographic and Health Survey (NDHS) reported anaemia rates of 58% among pregnant women and 68% among children under five. Nationally representative estimates for other nutrients, such as vitamin A, zinc, iodine, and iron deficiencies, are sparse; however, these deficiencies are still recognised as important public health concerns. Micronutrient deficiencies also contribute to other forms of malnutrition in Nigeria, where stunting affects 37% of children under five and wasting remains a persistent challenge, as reported in the 2018 NDHS.

Given the higher physiological requirements, pregnant women and infants are most vulnerable to these deficiencies [5]. Malnourished pregnant women face increased risks of adverse health outcomes, including low birth weight (LBW), preterm birth, and infant mortality [6,7,8]. They are also more susceptible to pregnancy-associated complications, such as hypertension, gestational diabetes, and postpartum haemorrhage [9], due to micronutrient deficiencies that disrupt normal metabolic, vascular, and immune functions. For example, an inadequate calcium and magnesium intake has been linked to hypertensive disorders in pregnancy, while a poor overall nutritional status can contribute to insulin resistance and abnormal glucose metabolism, increasing the risk of gestational diabetes [7]. Anaemia, due to iron deficiency, which contributes to around 50% of anaemia cases, can also raise the risk of a postpartum haemorrhage by impairing oxygen delivery and blood-clotting capacity [9,10]. Vitamin D and calcium deficiencies are the primary causes of rickets in children aged 3 to 24 months, leading to weakened bones, stunted growth, and delayed development [11,12]. Vitamin A deficiency (VAD) reduces resistance to infectious diseases and is the leading cause of preventable night blindness and, in severe cases, xerophthalmia and blindness [13]. Zinc deficiency is associated with adverse neonatal outcomes, including spontaneous abortions, low birth weight, and congenital malformations [14]. These deficiencies often coexist and are particularly critical during the first 1000 days of a child’s life, as an inadequate intake of these key nutrients increases the risk of lifelong health issues. While a global concern, these deficiencies disproportionately impact individuals in resource-poor settings, particularly in low- and middle-income countries (LMICs) in South Asia and sub-Saharan Africa [4]

Nigeria, the most populous country in sub-Saharan Africa (SSA) with over 214 million people across 36 states, faces significant challenges that are related to all forms of nutrition, including micronutrient malnutrition [15]. Despite having the largest economy in the SSA region, Nigeria is the African country with the highest number of people living below the extreme poverty line, and is one of the top 10 countries with children and pregnant women living with malnutrition [16]. Despite the significant public health importance of micronutrient deficiencies in Nigeria, there is no comprehensive synthesis of the prevalence rates and associated risk factors across its six geopolitical zones. Such information is crucial for informing targeted interventions, guiding policy decisions, and ensuring an equitable allocation of resources to address these regional disparities. Thus, this systematic review aims to address this gap.

## 2. Methods

This systematic review and meta-analysis were carried out using the preferred reporting items for systematic reviews and meta-analysis (PRISMA) [17]. The protocol was registered at the International Prospective Register of Systematic Reviews (PROSPERO) and can be accessed with the registration number: CRD42019124509.

### 2.1. Study Question

The authors used the Population, Exposure, Outcome (PEO) framework to include all relevant papers. The study question is: What is the prevalence of micronutrient deficiencies in pregnant women and children under the age of five years in Nigeria, and what are the associated risk factors for these deficiencies?

Population/participants: pregnant women and children under the age of five years in Nigeria.Exposure: participants with micronutrient deficiencies.Outcome: the prevalence and associated risk factors.

Micronutrient deficiency was defined based on the criteria outlined in the ESPEN micronutrient guidelines, and only studies that met these criteria were included [2].

Anaemia in pregnant women: a blood haemoglobin concentration of <11 g/dL or a haematocrit (Packed Cell Volume, PCV) less than 33%.Anaemia in children: a blood haemoglobin concentration of <12 g/dL and a PCV below 32%.Vitamin A deficiency: a serum retinol below 0.7 µmol/L.Vitamin C: a serum vitamin C below 0.5 mg/dL.Vitamin D: serum or plasma 25-hydroxyvitamin D levels below 75 nmol/L.Vitamin E: a serum concentration of less than 12 µmol/L.Magnesium: a serum magnesium that is level lower than 1.03 mEq/L.Calcium: serum calcium levels less than 8.5 mg/dL.Zinc: below the normal range of 130–140 µg/g for children below the age of five.

### 2.2. Search Strategies

We searched the databases PubMed, Cochrane Library, African Journals Online, and Global Health (EBSCO) for studies reporting on micronutrient deficiencies in pregnant women and children in Nigeria. Additional sources searched included Google Scholar, the author’s institutional library, conference proceedings, and the reference list of identified articles and reports. We used a combination of text words and MeSH (Medical Subject Headings) terms to conduct the searches as follows; (a) Population (child OR young child OR pregnant OR pregnant woman OR pregnancy) (b) Exposure (vitamin OR minerals OR nutrient OR anemia or anaemia OR magnesium OR calcium OR zinc) (c) Outcome (deficiency OR malnutrition OR deficient) AND (Nigeria or Nigeri*). These search terms were combined using the “OR” and “AND” Boolean operators. The original search was conducted in September 2018, and an updated search was conducted in July 2024 to identify any new studies. All studies that fit the review title were retrieved and screened for inclusion criteria.

### 2.3. Eligibility Criteria

We included all observational cross-sectional studies that were published in peer-reviewed journals between January 2008 and July 2024 that reported the prevalence of micronutrient deficiencies and the associated risk factors in pregnant women and children under five years old in Nigeria.

The included studies had the following characteristics:(i)Studies that involved pregnant women and children under the age of five in Nigeria.(ii)Studies that employed a cross-sectional design method and used standard criteria to define the deficiencies.(iii)Studies whose primary outcome was enumerating the prevalence of micronutrient deficiencies or related diseases, and studies that assessed the associated risk factors.(iv)Studies that were published in English.(v)Studies that were published between January 2008 and July 2024. Studies that did not meet these criteria were excluded.

### 2.4. Study Selection and Screening

The searches were downloaded into EndNote version X7 and de-duplicated. Then, screening and the selection of studies were conducted in two stages. First, the title and abstract screening were conducted, followed by a review of the full text. Through the title and abstract screening stage, studies that reported a prevalence of micronutrients of interest in pregnant women were selected for a full-text review. Then, from the full-text reviewing, any article that was classified as potentially eligible was considered as a full text and then screened against the inclusion criteria. At the time of screening, any disagreements were resolved through discussion.

### 2.5. Data Extraction

A standardised data extraction tool, implemented in Microsoft Excel, was used to extract the relevant study parameters based on the Joanna Briggs Institute data extraction format. For each study, the following data were extracted: (a) identification data (the first author’s last name and publication year); (b) the study’s region, geopolitical zone (GZ); (c) the study period; (d) the study design and sample size (N); (e) prevalence with 95% confidence intervals as the effect measure, and *p*-values for the associated risk factors (if available). All data were extracted independently and verified for accuracy before analysis.

### 2.6. Quality Appraisal and Risk of Bias

All the included studies were critically appraised for methodological quality using the ‘Guidelines for evaluating prevalence studies’ [18]. The included studies were rated based on three main domains: sampling, measurement, and analysis, which were divided into eight categories. For each category, one point was awarded if the answer was yes, and zero if the answer was no, resulting in a total of eight maximum points. Studies with scores of 0–2 were of low quality, 3–5 were moderate, and 6–8 were of high quality. The risk of bias for the selected studies was evaluated using the Risk of Bias tool [19], which assesses the internal and external validity based on study design-specific criteria with nine categories. Studies with 0–3 points were rated as low risk, 4–6 as medium risk, and 7–9 as high risk.

### 2.7. Data Synthesis and Analysis

We exported the data into RevMan version 5.4.1 for analysis and applied the random-effects model to estimate the pooled estimate of anaemia in pregnant women and children under five. Heterogeneity among the reported prevalences was assessed according to the square of the inverse variance (*I*^2^) test. Given this review, the *I*^2^ value was cut off at 50%, and a random effect model was applied to adjust for this observed variability and to determine the pooled effects. We generated forest plots with a 95% confidence interval (CI). A narrative synthesis was employed for other micronutrients of interest due to the limited number of available studies. The risk factors were narratively discussed and categorised based on the direction and statistical significance of their association with the outcome. Factors with a *p* < 0.05 were considered significant, while those with a *p* ≥ 0.05 were considered non-significant. The direction (positive or negative) of the association was also noted for all factors. Publication bias was assessed by visual inspection of the funnel plot and Egger’s regression test.

### 2.8. Ethical Consideration

Ethical consent was not considered, as the study was conducted using data extracted from published articles in research databases.

## 3. Results

### 3.1. Overview of the Search Results

As illustrated in Figure 1, the initial search of the various databases produced a total of 1207 records, from which 398 duplicate studies were removed. A further 750 were removed following stage 1, which included a screening of the titles and abstracts, and 63 articles were identified for full-text screening. In stage 2 screening, 26 studies that did not meet the inclusion criteria were excluded. Finally, 37 studies were selected and included in the systematic review: 27 for pregnant women and 10 for children under the age of five. For the meta-analysis, 26 studies were included in total: 19 studies were used to estimate the pooled prevalence of anaemia in pregnant women, and 7 studies were in young children. As other micronutrients of interest did not have sufficient data to pool, a narrative synthesis was presented for vitamins A, C, magnesium, zinc, and copper using data from 11 studies.

### 3.2. Characteristics of the Included Studies

This review included a total sample of 9889 pregnant women and 5642 children under five. The minimum sample size for pregnant women was 64 in a study conducted in Delta State in the South South region [20]. The maximum sample size was 1306 [21], and the participants were aged between 15 and 45. For young children, the sample size ranged from the smallest sample size of 44 [22] to the highest sample size of 2823 [23]. Table 1 provides a detailed view of the included studies.

### 3.3. Prevalence of Anaemia in Pregnant Women

As shown in Figure 2, the prevalence of anaemia among pregnant women in Nigeria varied widely across the included studies. The pooled prevalence, calculated using a random-effects model, was 54.0% (95% CI: 44.3%–62.9%; Z = 16.6; *p* < 0.001), indicating that more than half of pregnant women may be affected. The reported prevalence rates ranged from as low as 12.3% in Lagos State [39] to as high as 83.8% in another study from the same South West region [41]. This wide range highlights substantial variability in the study’s findings. Significant heterogeneity was observed across studies (*I*^2^ = 98.6%; *p* < 0.001), suggesting that differences in population characteristics, sampling methods, and regional factors likely contributed to the variation in prevalence estimates.

### 3.4. Prevalence of Other Micronutrient Deficiencies in Pregnant Women: Narrative Synthesis of Findings

Due to the limited number of eligible studies for vitamin A (*n* = 3), vitamin C (*n* = 2), vitamin E (*n* = 1), magnesium (*n* = 2), and copper (*n* = 2) deficiencies in pregnant women in Nigeria, a narrative synthesis was used to summarise their findings.

#### 3.4.1. Vitamin A Deficiency

Three studies assessed the prevalence of vitamin A deficiency in different regions of Nigeria, reporting rates that ranged from 34% to 65%. In the North Central region, prevalence ranged from 34% [42] to 65% [43]. In the South South region, a single study reported an intermediate prevalence of 48% [48]. The wide variation in prevalence may reflect regional differences in dietary intake, socioeconomic conditions, or access to supplementation programmes. However, variations in study design, sample characteristics, and diagnostic methods may also contribute to these discrepancies. Despite these differences, the findings collectively highlight vitamin A deficiency as a significant public health concern in Nigeria, warranting further investigation and targeted nutritional interventions, particularly in high-burden regions.

#### 3.4.2. Vitamin C and E Deficiencies

Only one study reported the prevalence of vitamin C and vitamin E deficiencies, both conducted in the North West region of Nigeria. There was a high prevalence of vitamin C deficiency at 80% [43] (95% CI: 0.76–0.83), while a subsequent study by the same research group reported a 51% prevalence of vitamin E deficiency (95% CI: 0.44–0.58) [47]. Although these figures suggest a substantial burden of deficiency, the limited number of studies, a single geographic location, and potential overlap in study populations restrict the generalisability of these findings. Moreover, both studies relied on small sample sizes and cross-sectional designs, limiting causal inference. The high prevalence rates, however, do align with broader evidence of micronutrient deficiencies in low-resource settings and highlight the need for expanded surveillance and targeted nutritional interventions. Further research is required across other regions to determine whether these deficiencies are widespread or geographically concentrated.

#### 3.4.3. Magnesium Deficiency

Two studies investigated magnesium deficiency among pregnant women in the South South region of Nigeria but reported markedly different prevalence rates. One study, conducted in an urban setting in Edo State, reported a relatively low prevalence of 16% among women assessed at 24–26 weeks’ gestation [44]. The sample also included teenage mothers, which may have influenced the nutritional status and health-seeking behaviours. In contrast, a much higher prevalence of 94% was reported in a rural area of neighbouring Delta State [20]. Their sample was largely composed of women aged 18–33, with 13% aged 38–49, and included participants from lower socioeconomic backgrounds. These demographic and geographic differences may help to explain the wide discrepancy in reported prevalence. Factors such as rural–urban location, maternal age, gestational stage, and socioeconomic status can influence the results.

#### 3.4.4. Copper Deficiency

Two studies assessed copper deficiency among pregnant women in the South East region of Nigeria, but reported sharply contrasting results. One study, conducted in an urban population in Enugu State, found no evidence of copper deficiency among participants [45]. However, Ugwuja et al. [46] reported a prevalence of 58.1% in a different population within the same region, specifically in Ebonyi State.

Two studies assessed copper deficiency among pregnant women in Nigeria’s South East region, but reported sharply contrasting results. One study in urban Enugu State found no evidence of copper deficiency [45]. However, Ugwuja et al. [46] reported a prevalence of 58.1% in Ebonyi State within the same region. The observed variation may reflect differences in the socioeconomic and nutritional status of the populations studied, as well as potential rural–urban disparities in dietary diversity and access to micronutrient-rich foods.

### 3.5. Prevalence of Micronutrient Deficiencies in Children

#### Prevalence of Anaemia in Children

The prevalence of anaemia among young Nigerian children varied substantially across the included studies (Figure 3). The lowest rate was reported at 10.3% in a study conducted in the South West [50], while the highest was 76.9% in the South East [23]. A pooled estimate using a restricted maximum likelihood (REML) random-effects meta-analysis model yielded a prevalence of 56% (95% CI: 35.3%–75.6%), suggesting that more than half of the young children in Nigeria may be affected by anaemia. The wide confidence interval and range in estimates reflect considerable heterogeneity across regions and study contexts. The following sections explore these potential risk factors in greater depth to better understand the underlying drivers of childhood anaemia in Nigeria.

### 3.6. Prevalence of Other Micronutrient Deficiencies in Young Children: Narrative Synthesis of Findings

Three studies reported on the prevalence of vitamin A, vitamin D, and zinc deficiencies among children under five in Nigeria, with the findings varying widely by nutrient and region. There was a relatively low prevalence of vitamin A deficiency (5.3%) in the South West [56], but a much higher prevalence of vitamin D deficiency (73.3%) in the North West [55]. This finding was also similar to another study conducted in the North Central region, which reported a high prevalence of zinc deficiency at 86.4% [22]. These findings suggest a considerable burden of micronutrient deficiencies, particularly in the northern parts of the country, which are historically associated with higher levels of poverty, food insecurity, and malnutrition. However, each of these estimates is based on a single study, limiting the generalisability of the findings. As such, these results should be interpreted with caution.

### 3.7. Risk Factors for Micronutrient Deficiencies in Pregnant Women and Children

Several risk factors for micronutrient deficiencies were consistently identified across the included studies, with some showing significant positive associations while others showed non-significant associations (Figure 4). These risk factors have been grouped into four key thematic areas, which are discussed below with supporting evidence from the included studies.

Dietary and Food Access

Low dietary iron intake, poor dietary quality, and household food insecurity were the most consistently reported risk factors, particularly among pregnant women and young children. All ten studies examining this theme reported significant associations between food insecurity and micronutrient deficiencies [34,38,40,41,46,48,49,50], making this the strongest and most consistent determinant across the evidence base.

2.Socioeconomic and Educational Status

Socioeconomic status was significantly associated with a micronutrient deficiency risk in nine out of ten studies, reinforcing the presence of a social gradient in nutritional health. Interestingly, two studies in urban areas observed that women from a higher socioeconomic status had increased odds of anaemia, potentially reflecting lifestyle factors or a nutrition transition, where economic development leads to shifts in dietary patterns that may lack essential micronutrients [25,27]. Associations with maternal education were mixed: seven out of eleven studies found that lower education was linked to a higher deficiency risk, while four studies found no significant relationship.

3.Health-Related Factors

A history of febrile illness, particularly malaria and diarrhoea, was significantly associated with micronutrient deficiencies in all seven studies assessing this factor. These illnesses were particularly impactful in children, with malaria showing the most substantial impact [23].

4.Reproductive and Demographic Factors

Age, parity, and interpregnancy intervals showed variable associations. Among women, an older age was linked to an increased deficiency risk in 11 out of 19 studies, while 8 studies found no significant effect. Boys and children under the age of three were at a higher risk of developing anaemia in three out of four studies, while girls were at a higher risk of developing zinc deficiency, which was assessed in one study. Parity also appeared influential: 10 out of 18 studies reported higher deficiency rates among women with greater parity, while 8 studies found no association. Short interpregnancy intervals were associated with an increased risk in seven out of nine studies. These findings suggest that both biological depletion and cumulative reproductive stress may influence micronutrient status [31,32,36,37,43,49,50,52].

## 4. Discussion

This systematic review and meta-analysis estimated the pooled prevalence of anaemia in pregnant women and children in Nigeria as 53.64% and 55.96%, respectively. Deficiencies in other micronutrients, such as vitamin A, vitamin D, zinc, magnesium, and copper, were also reported for both groups, with some studies indicating extremely high rates; for instance, a zinc deficiency was at 86.4% and a magnesium deficiency was at 94%. These pooled anaemia estimates are somewhat lower than the 2018 Nigeria Demographic and Health Survey (NDHS) figures of 58% in pregnant women and 68% in children under five. This difference may reflect variations in the study settings, as the included studies in this review were hospital-based and sub-national and may not capture rural or underserved populations with higher prevalences.

These findings highlight a significant burden and regional variability of micronutrient deficiencies among children under five and pregnant women in Nigeria. Compared with the data from neighbouring countries, the prevalence is higher. For example, anaemia prevalence in Uganda is 30% [57] and Ethiopia is 26.4% [58]; however, it is comparable to Sudan at 53% [59] and Ghana at 51% [60]. Among children, Ghana had an anaemia prevalence of 49.1% [61], while Kenya and Uganda reported 55% and 49%, respectively [62,63].

When compared to international benchmarks, the prevalence of anaemia and other micronutrient deficiencies in Nigeria exceeds the acceptable thresholds. According to the World Health Organization (WHO), anaemia prevalence above 40% in any population group is classified as a severe public health problem. While global efforts, such as the WHO Global Nutrition Targets 2025, aim to reduce anaemia in women of reproductive age by 50% and achieve a 40% reduction in stunting among children, Nigeria remains off track in meeting these targets, with minimal declines in anaemia rates over the past decade [15]. The persistently high burden of these deficiencies, particularly in northern regions and among low-income populations, suggests that the current interventions are either insufficient in scale or poorly targeted.

Anaemia is frequently caused by deficiencies in essential micronutrients, particularly iron, folate, and vitamin B12, which are required for effective erythropoiesis and haemoglobin function [64]. During pregnancy, the demand for these nutrients increases significantly due to the expansion of maternal blood volume and the nutritional needs of the growing foetus. The findings of this review reinforce this pathophysiological relationship, as high anaemia prevalence often coincides with other micronutrient deficiencies, including vitamin A, zinc, and magnesium. This supports that anaemia should not be addressed in isolation, but rather as part of a broader, multisectoral response to maternal undernutrition.

The risk factors identified point to a complex interplay of dietary, socioeconomic, health-related, and demographic factors that are driving the high prevalence of anaemia and micronutrient deficiencies among women and young children in Nigeria. These risk factors align with the United Nations Children’s Fund (UNICEF) conceptual framework on undernutrition, which highlights both immediate and underlying causes rooted in food access, care practices, and broader systemic inequities. Several studies have shown that women with lower dietary diversity scores are significantly more likely to experience anaemia and multiple micronutrient deficiencies [65,66]. For example, Uganda has recorded a decline from 41% to 32% in anaemia between 2006 and 2016, and this has been attributed to food fortification, dietary diversity, malaria control, and a strengthening of the health care system [67]. Among children, a low intake of iron, vitamins, and zinc-rich foods is a critical concern. Previous studies have reported that over 60% of Nigerian children consume less than the recommended daily intake (RDI) for key nutrients—a pattern often exacerbated by seasonal food shortages, high food costs, and food insecurity in the country [68].

Our analysis of associated risk factors reveals that socioeconomic and health-related inequalities significantly influence the maternal and child nutrition outcomes in Nigeria. Low household incomes and limited maternal education are consistently linked to poor dietary practices, reduced access to antenatal care, and an increased risk of micronutrient deficiencies, particularly in rural areas [69,70]. Infectious diseases, such as malaria, helminths, and diarrhoea, further exacerbate nutritional deficiencies by depleting micronutrient stores, especially in young children. Reproductive factors, such as early pregnancy, adolescent motherhood, and high parity, place additional physiological demands on women who often enter pregnancy with pre-existing nutritional deficits and limited support. These vulnerabilities contribute to poor maternal outcomes, neonatal complications, and low birth weights. As [7,71] emphasised, maternal undernutrition can perpetuate intergenerational cycles of stunting, anaemia, and poverty, and thus must be taken seriously.

Broader systemic challenges compound these individual-level risk factors. Poor infrastructure, weak health systems, and inadequate policy implementation continue to undermine nutrition programming in Nigeria. Although the country has adopted targeted initiatives, such as the National Policy on Food and Nutrition and school feeding programs, coverage gaps, weak intersectoral coordination, and implementation gaps constrain their impact. Strengthening health and nutrition surveillance, scaling up nutrition-sensitive interventions, and targeting high-risk populations are critical to addressing these deficiencies at scale.

This study is not without limitations. First, most of the included studies were sub-national and hospital-based, which limits the generalisability of the findings to the broader Nigerian population. Second, there was high heterogeneity across studies, likely due to the differences in sample populations, diagnostic methods, and settings, which may affect the pooled prevalence estimates. Third, there was a limited number of studies for certain micronutrients (e.g., magnesium, zinc, copper, and vitamins C and E), restricting the strength of the conclusions drawn about their prevalence and associated risk factors. However, the use of a random-effects model and the critical appraisal of included studies helped us to account for variability and enhance the robustness of the findings. Lastly, all studies included in this review used cross-sectional designs, limiting causal inference. The observed associations cannot establish temporal or causal relationships between risk factors and micronutrient deficiencies. Further longitudinal and interventional research is needed to clarify the causal pathways and support evidence-based policy responses in Nigeria.

## 5. Conclusions

This systematic review and meta-analysis revealed a high burden of anaemia among pregnant women and children in Nigeria, with at least one in two affected, spanning all geopolitical zones. The analysis also identified key associated risk factors, including poor dietary diversity, low maternal education, food insecurity, infectious diseases, and adverse reproductive health indicators, such as early pregnancy and high parity. Although this study did not assess the impacts on pregnancy outcomes directly, there is established evidence on the long-term consequences on maternal–foetal health [9]. Micronutrient malnutrition often results in a vicious intergenerational cycle of deficiencies, with significant implications for a population’s health and a country’s workforce productivity [1].

These findings highlight the need for comprehensive interventions that will address both the nutritional and wider determinants of health. Malnutrition is the result of multiple intersecting factors, including food insecurity, poverty, and social and political instability, which have persisted for decades in Nigeria. Consequently, an intersectoral approach, coordinating health, agriculture, education, and social protection sectors, is urgently needed. Policymakers should prioritise region-specific strategies that integrate micronutrient supplementation (such as routine iron–folic acid supplementation in pregnancy, the use of multiple micronutrient powders for young children, and large-scale food fortification), targeted nutrition education, improved antenatal care, and food security initiatives (such as conditional cash transfers and school feeding programmes). Strengthening health systems, expanding routine nutrition surveillance, and implementing national nutrition policies effectively will be critical to reducing the burden of micronutrient deficiencies and achieving improved maternal and child health outcomes across Nigeria.

## 6. Future Work

This review highlights critical gaps in the current evidence base on micronutrient deficiencies in Nigeria. Future research should prioritise large-scale, regionally representative studies. Particular attention is needed for under-researched micronutrients, such as magnesium, zinc, copper, and vitamins C and E, where data remain limited and inconsistent. To strengthen the policy and programme planning, there is an urgent need for nationally representative data on the prevalence and distribution of these understudied micronutrient deficiencies in Nigeria. Additionally, mixed-methods and community-informed research approaches are encouraged to capture the lived experiences, cultural dietary practices, and structural barriers influencing the nutritional outcomes. Such approaches would enrich the quantitative data and inform more context-sensitive interventions. Finally, there is an urgent need for rigorous evaluation of the existing nutrition programmes and supplementation initiatives to assess their real-world effectiveness.

## Figures and Tables

**Figure 1 nutrients-17-02338-f001:**
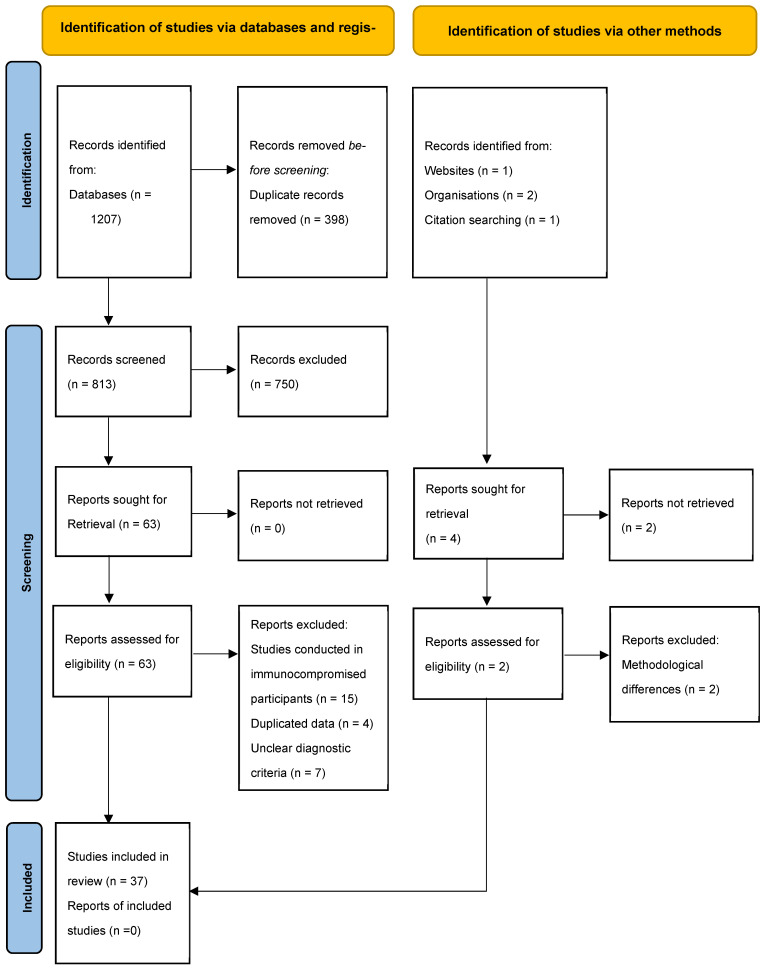
PRISMA 2020 flow diagram for new systematic reviews, which included searches of databases, registers, and other sources.

**Figure 2 nutrients-17-02338-f002:**
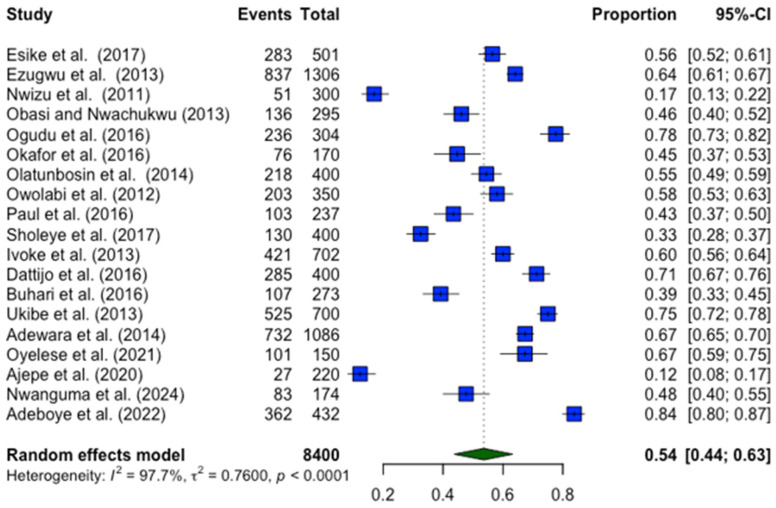
Forest plot of the results demonstrating the pooled prevalence of anaemia among pregnant women in Nigeria [21,24,25,26,27,28,29,30,31,32,33,34,35,36,37,38,39,40,41].

**Figure 3 nutrients-17-02338-f003:**
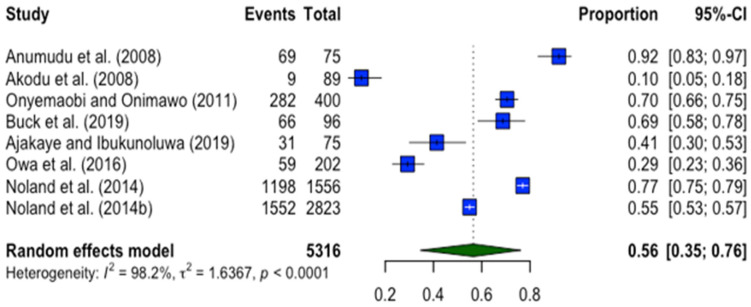
Forest plot of the results from a random-effects meta-analysis on the prevalence of anaemia in young Nigerian children. The results are presented as proportions with the corresponding 95% confidence interval [20,23,49,50,51,53,54].

**Figure 4 nutrients-17-02338-f004:**
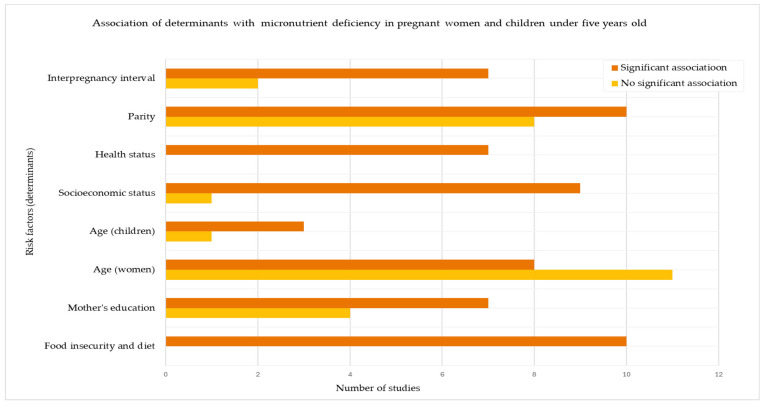
Direction of association between determinants and micronutrient deficiency. This horizontal bar chart illustrates the number of studies that reported either a significant or non-significant association between selected determinants and micronutrient deficiency. Each determinant is represented by two bars: one for studies showing a statistically significant association (e.g., a higher risk with food insecurity or low socioeconomic status) and one for those reporting no significant association.

**Table 1 nutrients-17-02338-t001:** Characteristics of the studies included in this review.

S/N	Author	State	GZ	Sample Size	Indication	Prevalence	Quality	Risk of Bias
Studies conducted with pregnant women								
1. ***​***​	Adewara et al. [24]	Niger	NC	1086	Anaemia	67.4%	High	Low
2. ***​***​	Paul et al. [25]	Jos	NC	237	Anaemia	43.5%	High	Low
3. ***​***​	Esike et al. [26]	Ebonyi	SE	501	Anaemia	56.5%	High	Low
4. ***​***​	Ezugwu et al. [21]	Enugu	SE	1306	Anaemia	64.1%	High	Low
5. ***​***​	Obasi and Nwachukwu [27]	Ebonyi	SE	295	Anaemia	45.7%	High	Low
6. ***​***​	Nwanguma et al. [28]	Enugu	SE	174	Anaemia	47.7%	High	Low
7. ***​***​	Ukibe et al. [29]	Anambra	SE	700	Anaemia	75.0%	Moderate	Low
8. ***​***​	Ogudu et al. [30]	Ebonyi	SE	304	Anaemia	77.9%	High	Low
9. ***​***​	Nwizu et al. [31]	Kano	NW	300	Anaemia	17.0%	High	Low
10. ***​***​	Buhari et al. [32]	Sokoto	NW	273	Anaemia	39.2%	High	Low
11. ***​***​	Dattijo et al. [33]	Bauchi	NE	400	Anaemia	71.3%	High	Low
12. ***​***​	Okafor et al. [34]	Cross river	SS	170	Anaemia	44.7%	High	Low
13. ***​***​	Olatunbosun et al. [35]	Uyo	SS	400	Anaemia	54.5%	High	Low
14. ***​***​	Ivoke et al. [36]	Ebonyi	SW	702	Anaemia	60.0%	High	Low
15. ***​***​	Owolabi et al. [37]	Oyo	SW	350	Anaemia	58.0%	High	Low
16. ***​***​	Oyelese et al. [38]	Ogun	SW	150	Anaemia	67.3%	High	Low
17. ***​***​	Ajepe et al. [39]	Lagos	SW	220	Anaemia	12.3%	High	Low
18. ***​***​	Sholeye et al. [40]	Ogun	SW	400	Anaemia	32.5%	High	Low
19. ***​***​	Adeboye et al. [41]	Lagos	SW	432	Anaemia	83.4%	High	Low
20. ***​***​	Hanson et al. [42]	Abuja FCT	NC	87	Vitamin A	35.0%	High	Low
21. ***​***​	Ugwa [43]	Abuja, FCT	NC	200	Vitamin A	Vit A: 65.0% Vit E: 51.0%	Moderate	Medium
22. ***​***​	Enaruna et al. [44]	Edo	SS	160	Magnesium	16.3%	High	Low
23. ***​***​	Osadolor and Omogiade [20]	Delta	SS	64	Magnesium	93.8%	High	Medium
24. ***​***​	Nwagha et al. [45]	Enugu	SE	130	Copper	0%	High	Low
25. ***​***​	Ugwuja et al. [46]	Ebonyi	SE	349	Copper	Cu: 58.2% Zn:45.8%	High	Low
26. ***​***​	Ugwa et al. [47]	Abuja, FCT	NC	400	Vitamin C	79.5%	Moderate	Medium
27. ***​***​	Williams et al. [48]	Calabar	SS	99	Vitamin A	48.5%	High	Low
Studies conducted with young children								
28. ***​***​	Anumudu et al. [49]	Ogun	SW	75	Anaemia	92%	Moderate	Low
29. ***​***​	Akodu et al. [50]	Lagos	SW	89	Anaemia	10.1%	High	Low
30. ***​***​	Owa et al. [51]	Oyo	SW	202	Anaemia	29.2%	High	Low
31. ***​***​	Onyemaobi and Onimawo [52]	Imo	SE	400	Anaemia	70.5%	High	Low
32. ***​***​	Buck et al. [53]	Imo	SE	96	Anaemia	69%	High	Low
33. ***​***​	Ajakaye and Ibukunoluwa [54]	Edo	SE	75	Anaemia	41.3%	High	Low
34. ***​***​	Noland et al. [23]	Abia Plateau	SE NC	1556 2823	Anaemia	76.9% 54.9%	High	Low
35. ***​***​	Akeredolu et al. [55]	Zaria	NW	112	Vitamin D	73.3%	High	Low
36. ***​***​	Abolurin et al. [56]	Osun	SW	170	Vitamin A	5.3%	High	Low
37. ***​***​	Jaryum et al. [22]	Plateau	NC	44	Zinc	86.4%	Moderate	Medium

GZ: Geopolitical zone; NC: North Central, NW: North West, NE: North East, SS: South South, SE: South East, SW: South West, FCT: Federal Capital Territory.

## Data Availability

The data that support the findings of this study are available via reasonable request to the corresponding author.

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
