# Peer review of "Micronutrient Deficiencies and Determinants Among Pregnant Women and Children in Nigeria: Systematic Review and Meta-Analysis"

_nutrients, 2025, doi:10.3390/nu17142338_

Round 1
Reviewer 1 Report
Comments and Suggestions for Authors
Micronutrient deficiencies among vulnerable communities in Nigeria are a significant and current public health issue that your paper tackles. You provide a comprehensive meta-analysis and systematic review of research conducted from 2008 to 2024 that assesses the prevalence and contributing factors of deficits in children under five and pregnant women. The paper is methodologically sound, well-structured, and extremely pertinent to guiding health and policy initiatives in low- and middle-income (LMIC) nations.
Your study's strengths include the topic's high relevance to maternal and child health worldwide, especially in LMICs, and the fact that it offers up-to-date pooled estimates and risk variables unique to Nigeria—a crucial consideration given the nation's sizable and diverse population.
Transparency and reproducibility are improved by using PRISMA rules and PROSPERO registration; regional inclusivity is ensured by a thorough search strategy across several databases, including African Journals Online; and consistency is added by using ESPEN definitions and unambiguous eligibility requirements.
Random-effects models are used appropriately in this well-executed meta-analysis of anemia prevalence. Bias risk and quality evaluation are thoroughly explained and supported by validated instruments. It makes sense to apply narrative synthesis to micronutrients for which there is little evidence. The comprehensive and insightful identification of the main determinants—dietary, socioeconomic, health-related, and reproductive—is commendable. Results are discussed in relation to national and international nutrition policy objectives.
Ideas for Improvement
1. Limited national representativeness: the majority of research is sub-national and facility-based. Although you recognize this drawback, a more explicit request for nationally representative data gathering would strengthen the conclusion.
2. Data imbalance across micronutrients: For managing reader expectations, note in the abstract that certain minerals (such as magnesium, copper, and vitamins C and E) are underrepresented with only one or two studies.
3. Absence of gender disaggregation in children: It is not stated in the analysis for children under five whether data were analyzed for variations in prevalence or risk depending on gender. This could improve the findings' nuance.
4. A few small clarifications are required: Figure/Table insertions: Before final publishing, statements like "[INSERT Table 1 HERE]" and "[INSERT Figure 4 HERE]" must be revised. Definitions of Acronyms: Some abbreviations, like REML, are not explained in the main text, which could make it more difficult for readers in general to understand; Type-related problems: There are a few minor issues, such irregular formatting (e.g., duplicated information in the author contact area and spacing around citations).
5. Policy implications: Although the topic of policy significance is covered, the conclusion would benefit from more specific, doable suggestions (such as models for intersectoral collaboration or particular micronutrient supplementation techniques).
Author Response
Comment 1: Limited national representativeness: the majority of research is sub-national and facility-based. Although you recognize this drawback, a more explicit request for nationally representative data gathering would strengthen the conclusion.
Response 1: This has now been added on page 16, lines 468-470.
there is an urgent need for nationally representative data on the prevalence and distribution of these understudied micronutrient deficiencies in Nigeria.
Comment 2: Data imbalance across micronutrients: For managing reader expectations, note in the abstract that certain minerals (such as magnesium, copper, and vitamins C and E) are underrepresented with only one or two studies.
Response 2: We have now added on page 1, lines 21-22.
Narrative synthesis was conducted to synthesise evidence on other micronutrients (Magnesium, copper and vitamins C and E) due to limited data
Comment 3: Absence of gender disaggregation in children: It is not stated in the analysis for children under five whether data were analyzed for variations in prevalence or risk depending on gender. This could improve the findings' nuance.
Response 3: Thank you for this comment. We have now added on page 12, lines 338-342.
Age, parity, and interpregnancy intervals showed variable associations. Among women, older age was linked to with increased deficiency risk in 11 out of 19 studies, while 8 found no significant effect. Boys and children under the age of three were at higher risk of developing anaemia in 3 out of 4 studies while girls were at a higher risk of developing zinc deficiency, this was assessed in one study.
Comment 4: A few small clarifications are required: Figure/Table insertions: Before final publishing, statements like "[INSERT Table 1 HERE]" and "[INSERT Figure 4 HERE]" must be revised. Definitions of Acronyms: Some abbreviations, like REML, are not explained in the main text, which could make it more difficult for readers in general to understand; Type-related problems: There are a few minor issues, such irregular formatting (e.g., duplicated information in the author contact area and spacing around citations).
Response 4:
- Figure/Table insertions have now been corrected throughout the manuscript.
- REML abbreviation has now been amended by including the full meaning on page 11, line 288.
- Duplicated information in the author contact has now been amended on page 1, lines 6-9. Spacing around citations have been fixed throughout the manuscript.
Comment 5: Policy implications: Although the topic of policy significance is covered, the conclusion would benefit from more specific, doable suggestions (such as models for intersectoral collaboration or particular micronutrient supplementation techniques).
Response 5: Thank you for this comment. We have now added on pages 15-16, lines 451 to 459, the below statement:
Malnutrition is the result of multiple intersecting factors, including food insecurity, poverty, and social and political instability, which have persisted for decades in Nigeria. Consequently, an intersectoral approach, coordinating health, agriculture, education, and social protection sectors is urgently needed. Policymakers should prioritise region-specific strategies that integrate micronutrient supplementation (such as routine iron–folic acid supplementation in pregnancy, use of multiple micronutrient powders for young children, and large-scale food fortification), targeted nutrition education, improved antenatal care, and food security initiatives (such as conditional cash transfers and school feeding programmes).
Reviewer 2 Report
Comments and Suggestions for Authors
This is an interesting systematic review and meta-analysis with sufficient novelty. Some points should be addressed.
- In Abstract, lines 17 and 21, the stetement "children under five" should be "children under five years old".
- At the end of the 1st paragraph of Introduction section, the authors should report previous epidemiological data for Nigeria concerning the nutrients deficiencies in this country.
- The sentence in lines 55-56 "They are also more susceptible to pregnancy-associated complications such as hypertension, gestational diabetes, and postpartum haemorrhage" needs more analysis.
- In the last paragraph of the Introduction, the authors should emphasize the literature gap that their systematic review will cover.
- In line 76, please revise as "children under the age of five years in Nigeria". The same should be revised at the following text (e.g. children under five years).
- The sentence in lines 122-125 is quite complex and needs revision concerning its syntax.
- In Figure 1, some numbers of studies into boxes are missing (e.g. Studies included in review (n=?).
- In Table 1, the year of publication (2nd column) could be omitted.
- the statements in lines 227-231 is quite complex, and confusing and needs revision concerning the English language grammar.
- In line 268, the statement "In contrast, [31] reported..." should be revised.
- Figure 4 should be cited into the text. Moreover, more analysis describing Figure 4 should be added into the text.
- The sentence in lines 258-262 is quite complex and it should be split into two lower sentences.
- If the studies including in this systematic review are mainly cross-sectional, then the authors should report this issue in the limitations of their study, as the cross-sectional design of studies cannot support causality effects.
- English language editing is highly recommended.
English language editing is highly recommended.
Author Response
Comment 1: In Abstract, lines 17 and 21, the statement "children under five" should be "children under five years old".
Response 1: Thank you for this comment, we have now changed the statement to children under five years old across the manuscript.
Comment 2: At the end of the 1st paragraph of Introduction section, the authors should report previous epidemiological data for Nigeria concerning the nutrients deficiencies in this country.
Response 2: We have added the below statement on page 2, lines 44-51.
In Nigeria, micronutrient deficiencies remain a significant public health problem. The 2018 Nigeria Demographic and Health Survey (NDHS) reported anaemia rates of 58% among pregnant women and 68% among children under five. Nationally representative estimates for other nutrients, such as vitamin A, zinc, iodine, and iron deficiencies, are sparse; however, these deficiencies are still recognised as important public health concerns. Micronutrient deficiencies also contribute to other forms of malnutrition in Nigeria, where stunting affects 37% of children under five and wasting remains a persistent challenge, as reported in the 2018 NDHS.
Comment 3: The sentence in lines 55-56, "They are also more susceptible to pregnancy-associated complications such as hypertension, gestational diabetes, and postpartum haemorrhage" needs more analysis.
Response 3: Thank you for this we have addressed this by expanding the discussion, we have included the statement below on page 2, lines 56-61.
They are also more susceptible to pregnancy-associated complications such as hypertension, gestational diabetes, and postpartum haemorrhage [9], due to micronutrient deficiencies that disrupt normal metabolic, vascular, and immune functions. For example, inadequate calcium and magnesium intake has been linked to hypertensive disorders in pregnancy, while poor overall nutritional status can contribute to insulin resistance and abnormal glucose metabolism, increasing the risk of gestational diabetes [7].
Comment 4: In the last paragraph of the Introduction, the authors should emphasize the literature gap that their systematic review will cover.
Response 4: Thank you, we have addressed this on page 2, lines 80-85 by including this statement.
Despite the significant public health importance of micronutrient deficiencies in Nigeria, there is no comprehensive synthesis of prevalence rates and associated risk factors across its six geopolitical zones. Such information is crucial for informing targeted interventions, guiding policy decisions, and ensuring equitable allocation of resources to address regional disparities. Thus, this systematic review aims to address this gap.
Comment 5: In line 76, please revise as "children under the age of five years in Nigeria". The same should be revised at the following text (e.g. children under five years).
Response 5: This has been addressed on page 3 in lines 95 and 97.
Comment 6: The sentence in lines 122-125 is quite complex and needs revision concerning its syntax.
Response 6: Thank you for this. We have addressed the syntax error in lines 131-134 on page 3.
We included all observational cross-sectional studies that were published in peer-reviewed journals between January 2008 and July 2024, that reported the prevalence of micronutrient deficiencies and associated risk factors in pregnant women and children under five years old in Nigeria.
Comment 7: In Figure 1, some numbers of studies into boxes are missing (e.g. Studies included in review (n=?).
Response 7: Thank you for this. We have amended the figures to reveal the numbers in Figure 1, page 6
Comment 8: In Table 1, the year of publication (2nd column) could be omitted.
Response 8: We have removed the years in column 2 of table 1 on page 8.
Comment 9: the statements in lines 227-231 is quite complex, and confusing and needs revision concerning the English language grammar.
Response 9: we have revised this statement on pages 9-10, lines 235-238.
Three studies assessed the prevalence of vitamin A deficiency in different regions of Nigeria, reporting rates ranging from 34% to 65%. In the North Central region, prevalence ranged from 34% [25] to 65% [26]. In the South South region, a single study reported an intermediate prevalence of 48% [27].
Comment 10: In line 268, the statement "In contrast, [31] reported..." should be revised.
Response 10: We have revised this statement on page 10, lines 277-279 as
One study in urban Enugu State found no evidence of copper deficiency [30]. However, Ugwuja et al. [31] reported a prevalence of 58.1% in Ebonyi State within the same region.
Comment 11: Figure 4 should be cited into the text. Moreover, more analysis describing Figure 4 should be added into the text.
Response 11: We have cited figure 4 more prominently on lines 314, page 12 and an analysis is provided in section 3.7 on pages 11-12, lines 313-316.
Several risk factors for micronutrient deficiencies were consistently identified across the included studies, with some showing significant positive association and others non-significant associations (Figure 4). These risk factors have been grouped into four key thematic areas, which are discussed below with supporting evidence from the included studies.
Comment 12: The sentence in lines 258-262 is quite complex and it should be split into two lower sentences.
Response 12: We have addressed this on page 10, lines 265-269, we have split into two sentences as -
Their sample was largely composed of women aged 18–33, with 13% aged 38–49, and included participants from lower socioeconomic backgrounds. These demographic and geographic differences may help explain the wide discrepancy in reported prevalence. Factors such as rural–urban location, maternal age, gestational stage, and socioeconomic status can influence results.
Comment 13: If the studies including in this systematic review are mainly cross-sectional, then the authors should report this issue in the limitations of their study, as the cross-sectional design of studies cannot support causality effects.
Response 13: Thank you for this, we have included this limitation on page 15, lines 435-438.
Lastly, all studies included in this review used cross-sectional designs, limiting causal inference. Observed associations cannot establish temporal or causal relationships between risk factors and micronutrient deficiencies.
Comment 14: English language editing is highly recommended.
Response 14: We have edited the manuscript for language errors.
Round 2
Reviewer 2 Report
Comments and Suggestions for Authors
The authors have considerably revised and improved their manuscript.